# Hybrid Wind/PV E-Bike Charging Station: Comparison of Onshore and Offshore Systems

**Wardah Afzal, Li-Ye Zhao** **, Guang-Zhi Chen and Yu Xue \***

College of Engineering, Ocean University of China, Qingdao 266100, China; wardahafzal@gmail.com (W.A.);
18806391261@163.com (L.-Y.Z.); chenchen20210801@163.com (G.-Z.C.)
\* Correspondence: xueyu7231@ouc.edu.cn

**Abstract:** The concept behind this research article is advancement towards utilizing renewable energy sources of wind–solar to generate electrical energy for E-bike (electric bike) charging stations. To optimize the design and operation control of the wind–solar E-bike charging station system, the development of modelling this hybrid power generation system, consisting of solar and wind energy combined with battery storage, is proposed and will be studied in this paper. A university campus setting is utilized for the case study by comparing offshore (Huangdao) and onshore (Laoshan) sites. The proposed research will focus on annual energy production (AEP) and system cost analysis. The proposed work's main objectives are to analyze the wind/solar properties of the installation's location using the last 20 years' data, calculate the AEP for wind turbines and solar PV, and estimate how many E-bikes can be charged day/year with reliable operation. We have calculated that the hybrid power available is 27.08 kWh/day offshore and 22 kWh/day onshore. This research study concludes that on average, based on AEP, in the case of offshore, 5110 E-bikes can be charged per year and in the case of onshore, 4015 E-bikes can be charged per year. We have also calculated the COE (cost of energy) for 20 years for the proposed project, which is $0.62/kWh onshore and $0.46/kWh offshore.

**Keywords:** E-bike; hybrid wind/PV charging station; wind turbine; solar energy; system simulation; AEP calculation; COE

## 1. Introduction

Global utilization of electric vehicles and electric bikes has greatly expanded in recent years. However, because more electric vehicles will be on the road, there is a compelling need to create effective infrastructures to handle their increased numbers. To meet customers' demands for a dependable electric supply, charging station accessibility and positioning are also becoming increasingly crucial. Several nations have already started building these infrastructures. For instance, according to the UK Department for Environment, the British government intends to outlaw the sale of traditional diesel and gasoline cars and vans by 2040 [1].

Electric bikes (E-bike) are gaining popularity due to their lower emissions and less dependence on fossil fuels. Using energy from renewable sources inside distribution networks and the electrification of charging stations using smart grids offer a solution that enhances power conversion efficiency and lowers emissions [2]. Mainly, energy storage systems and distributed energy resources are utilized to power microgrids that can operate in grid-connected or island modes and support a range of loads [3]. Although the increasing number of high-capacity E-bike charging stations increases the demand for infrastructure, this puts strain on the electrical grid [4]. To address the challenges associated with power demand, regionally generated power from renewable energy sources is incorporated using a suitable power converter architecture [5]. Manufacturers of electric vehicles like Tesla (using Solar City) and Nissan Leaf (using Sun Power) provide charging station services as part of their efforts to expand the infrastructure for charging [6]. However, using renewable

energy in charging stations enhances utility grid synchronization while reducing charging costs and emissions [7,8].

We must charge E-bikes using sustainable electricity sources, including solar or wind energy, if they are to be sustainable. Clean and ecologically friendly energy sources include solar and wind power. Their inconsistent power generation, however, poses a risk to the electrical grid's stability. As a result, adding storage is a practical way to balance the supply and demand of power by storing excess energy and producing it as needed. Thus, a hybrid system that is both safer and more ecologically friendly is produced [9].

Maximum power point tracking (MPPT) techniques are used to obtain the ideal load power when the wind speed changes or the solar irradiation declines. A hybrid solar–wind turbine system can satisfy the load requirement as the basis for an uninterrupted supply. Power demand may be met continuously by combining solar PV and wind power in a single-generation system. This research study studies the design of a hybrid E-bike charging station powered by wind and solar energy [10]. The charging station features integrated battery storage, enabling it to operate on and off the grid. According to a research project termed a modelling study of a hybrid wind–solar electric bike charging station, it is feasible to set up a charging station for electric bikes that uses both wind and solar power [11]. As part of the study, a mathematical model was created that replicates the functioning of the charging station under various conditions and scenarios. Researchers can test various operational scenarios, look into various design possibilities, and enhance the performance of the charging station by using a modelling technique and critically analyzing the location for installation of such projects. The study might offer ways to overcome specific difficulties and restrictions [12].

Increasing awareness of the environmental consequences of fossil fuels has generated a strong drive towards transitioning to cleaner energy sources. Renewable energies, particularly wind and solar power, have emerged as viable alternatives due to their widespread availability and ability to provide sustainable energy without the drawbacks of fossil fuels. Furthermore, there has been a notable surge in interest surrounding the utilization of hydrogen as a clean substitute for traditional fossil fuels [13].

Overall, this research study underlines the need for additional research and development in this field and offers insightful information about the performance and potential of hybrid wind and PV E-bike charging stations. China's coastline's vastness, variety, and length make it a particularly interesting location to study renewable energy sources. The complementarity and long-term variability of solar and wind energy, particularly in the context of land–sea coordination development are, however, yet largely investigated [14,15]. Table 1 outlines the category, focus, and techniques of earlier studies on China's potential for renewable energy, and it makes this research gap clear. The survey results demonstrate that academics are disposed to independently investigate the solar/wind energy possibilities at onshore or offshore sites in China [16]. A modelling study on a hybrid wind–solar electric bike charging station is a novel and important research issue that can potentially promote sustainable transportation systems' development. The main objectives of this research study are the following:

- To analyze the geographical location for the installation;
- Calculate the AEP (annual energy production) for wind turbine and solar PV (photovoltaic);
- Comparison of two different offshore and onshore sites;
- Modelling of wind–solar charging station on MATLAB/Simulink;
- Calculation of E-bike charging for both locations;
- To do the cost analysis of hybrid energy using COE.

The rest of this article is divided into the following sections: essential factors in building E-bike charging stations are discussed in Section 2, along with designing hybrid E-bike charging stations. The methodology and analysis of results of E-bike charging systems using MATLAB Simulink is described in Section 3. The location analysis, cost analysis, and comparison between offshore and onshore using AEP and COE are covered

in Sections 4 and 5. The future scope and conclusion and recommendations of the proposed research is discussed in Sections 6 and 7.

**Table 1.** Comparison of different research studies.

| Reference | Objective | Methods | Findings |
|---|---|---|---|
| [13] | RES power production potential assessment | Yearly average solar radiation and wind speed data | Assess theoretical global solar and wind potential with constraints |
| [10] | Solar photovoltaic potential assessment | Geospatial estimation in 66 countries using the Belt and Road Initiative region | Evaluate the technical potential of solar photovoltaic power. |
| [11] | Solar power variability assessment | Multi-year hourly meteorological time-series data | Quantify spatiotemporal trends of global solar radiation and PV power in China. |
| [16] | Wind energy trends assessment | Analysis of annual and seasonal wind speed and energy resources | Significant decreasing trend in wind speed and energy resources in China |
| [14] | Optimal hybrid renewable energy system for Moroccan supermarkets with EV charging | Proposed PV/wind/battery system using HOMER grid software; analyzed three Moroccan cities | Dakhla site best with 71.66% renewable energy and low costs; sensitivity analysis performed |
| [15] | Robust EV charging station location optimization in micro-grids with renewables | Developed model considering EV demand and renewable uncertainty; used load fluctuation rate and kernel density estimation | Model demonstrated robustness and cost-effectiveness in simulations on IEEE 33-node network |
| [17] | Solar and wind potential assessment | Multi-year hourly satellite-derived MERRA-2 data | Analyze solar PV and wind resource potential and variability in India |
| [18] | Global wind energy trends assessment | Assessment of inter-annual variability and trends in wind energy | Identify significant trends in wind energy generation across countries |
| [19] | Long-term trend estimation | Accurate estimation of solar and wind energy potential | Importance of site-specific estimation for renewable energy expansion and resilience |
| [20] | Review wind-solar HRES for system modelling and optimization. | Analyzing models, HESS, converters, and algorithms while reviewing the literature. | Research in enhancing the performance of wind-solar HRES |
| [21] | Integrating PV-HESS with a multi-port DC-DC converter | PV-battery-EV charging integration used a bidirectional converter with a PI controller. | Enhanced integration, efficiency, and cost-effectiveness, mitigating overload issues during EV charging. |
| [22,23] | Solar–wind energy characterization | Hourly time-series meteorological data for onshore and offshore sites | Explore spatiotemporal variability and complementarity of solar and wind energy potential in China |

## 2. Essential Factors in Building E-Bike Charging Station

E-bikes and electric scooters offer a practical way to commute within cities. They have many advantages, including door-to-door connection, low (indirect) emissions, lessened parking and traffic jams, and a fraction of the energy consumption of an electric car. About 30% of the 928,000 bikes sold in the Netherlands in 2016 were electric bikes. Using a motor of up to 250 W (up to 1000 W for high-speed E-bikes) and a 12–48 V battery with an energy capacity of 0.2–1 kWh, an E-bike can travel up to 25 km/h. On the other hand, electric mopeds (including speed pedelec) may travel up to 45 km/h with a motor of 1–4 kW and normally require a 48 V battery of 1–5 kWh [15]. This is significantly less than an electric car's energy consumption of 150–200 Wh/km. E-bikes can be converted into a completely sustainable mode of transportation for daily commuting by installing a charging station at the workplace and running it on solar power [14].

The building of an E-bike charging station at the university that uses a combination of wind and solar energy is the primary focus of the proposed research. Developing an E-bike charging station requires careful consideration of several essential factors to

guarantee that the facility is practical, accessible, and efficient. One of the important aspects is location. The location of the charging station should be convenient for electric bike riders. The location should have enough space to build parking areas for electric bicycles and charging equipment. The facility should preferably be close to busy areas, such as university, supermarkets, etc. The cost as well as the sustainability of the charging station must be taken into account. The station must be designed with the least amount of energy and impact on the environment and be operationally and financially viable [24]. Considering these important factors, it is possible to design and build an E-bike charging station that meets user needs, is reasonably priced, environment friendly, and valuable [25]. Figure 1a below shows the suggested hybrid wind and PV E-bike charging station design for universities, shopping centers, or public transportation hubs to guarantee that E-bike riders can utilize it quickly. A real hybrid wind/PV system for lighting up street lights has been installed by Sunning Solar Company in China, as seen in Figure 1b, as described in reference [26]. The development of a hybrid wind/PV system to install an E-bike charging station specifically designed to meet the demands of university students was inspired by this successful implementation, and we used the knowledge obtained to do so.

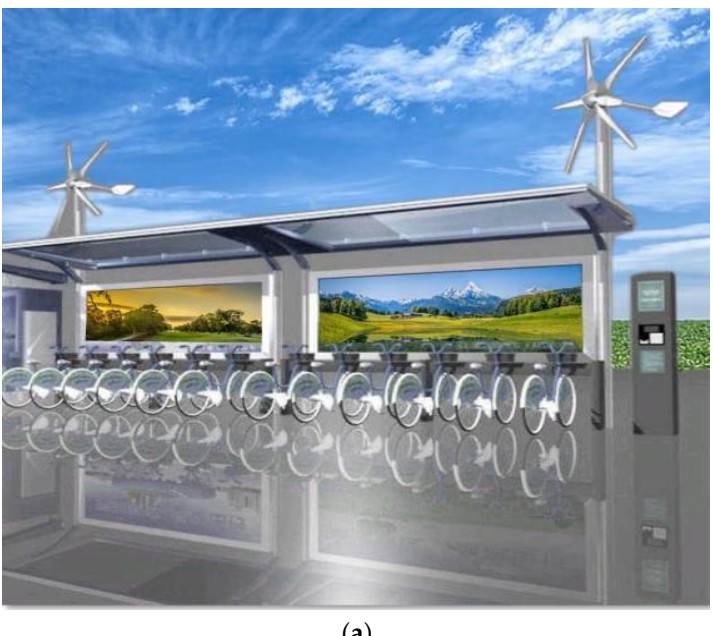

(**a**)

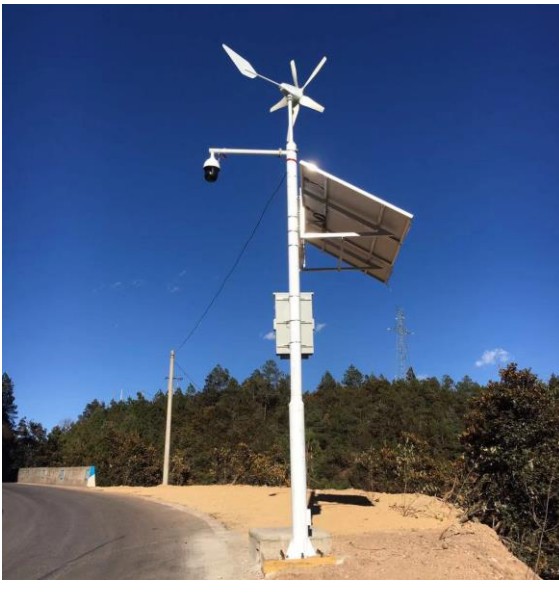

(**b**)

**Figure 1.** (**a**) Proposed design of hybrid wind/PV E-bike charging station. (**b**) Real hybrid wind/PV system [26].

*Designing Hybrid E-Bike Charging Station*

The sizing methodology for a solar–wind hybrid EV charging station encompasses the following steps (See Figure 2):

The design aspects of a charging station for electric vehicles are influenced by the E-bikes' characteristics and the energy source used. When designing a charging station, the following parameters need to be taken into consideration:

- Number of E-bike that can be charged: The charging station should be constructed to support the maximum number of vehicles charged concurrently or within a specific amount of time.
- Charge time: The charging station needs the proper infrastructure to satisfy the necessary charge time, whether for fast or slow charging.
- Charging connectors: To assure compatibility with the charging of electric vehicles, the charging station should offer relevant sockets or connectors. Examples that are frequently used are Type 1 (J1772), Type 2 (Mennekes), and CHAdeMO.

- Battery type and capacity: There are various EV models with various battery kinds and capacities. The charging station should meet the capacity needs of different battery types, including lithium-ion, nickel-metal hydride, and solid-state batteries.
- Potential of energy sources: When planning the charging station, it is essential to consider the potential and accessibility of energy sources such as grid electricity, solar power, and wind power. This will aid in determining the potential sources of energy and the necessary capacity.
- Dimensions of the station: Based on the available space and anticipated customer demand, the physical dimensions of the charging station, including the space needed for charging infrastructure, parking spaces, and any other facilities, should be decided.

Considering these parameters ensures that the charging station is designed to meet the specific requirements of the EVs and the available energy sources, enabling efficient and effective charging operation. Table 2 explains the different types of charging methods, their charging times, and applications.

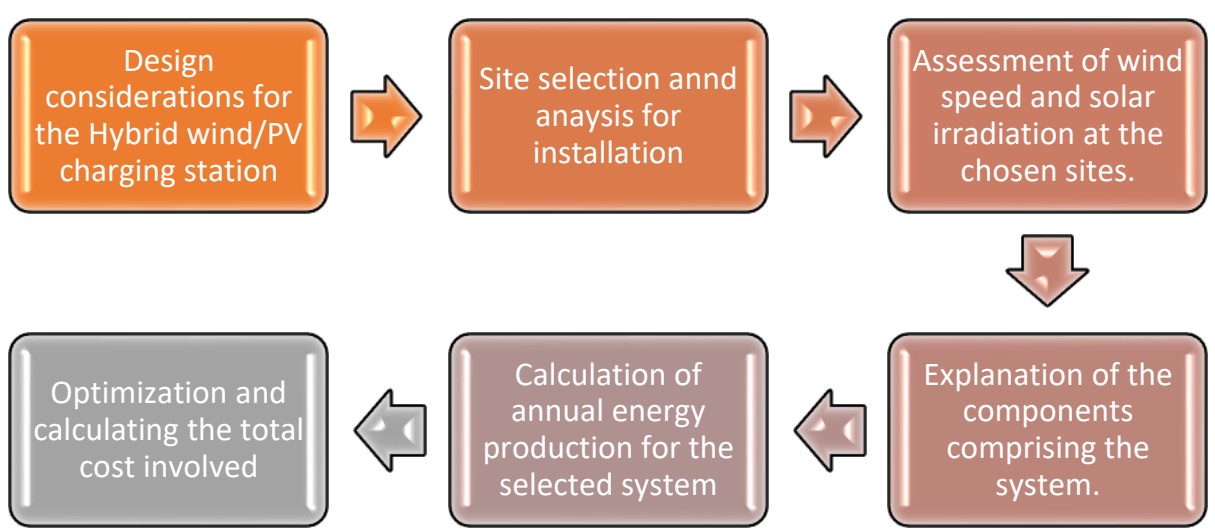

**Figure 2.** Steps for a hybrid wind/solar EV charging station.

**Table 2.** Methods to charge E-bike.

| Charging Method | Charging Level | Charging Speed (Range per Hour) | Equipment | Suitable for |
|---|---|---|---|---|
| Level 1 charging | Slow | 3–5 miles | Standard household outlet | Overnight charging and emergency top-ups (PHEVs) |
| Level 2 charging | Medium | 10–30 miles | 240-volt AC charging station | Daily charging for most BEVs and PHEVs |
| DC fast charging | Fast | Up to 60–80 miles (in 20 min) | High-power DC fast charger | Long-distance travel and quick top-ups on the road |

## 3. Methodology

In order to determine the effects of various factors, including wind speed and sun irradiation, on the functioning of the charging station, the study also performed a sensitivity analysis. The sensitivity analysis revealed that wind speed and solar irradiance substantially impacted the charging station's performance and that adjusting these variables might dramatically boost the station's efficiency. The suggested hybrid wind–solar electric bike charging station with the innovative DC–DC converter topology has the potential to offer a cost-efficient and sustainable solution for electric bike charging infrastructure, according to the study's findings [17]. The methodology described in this study [27], which defines a methodical strategy for the management of energy storage and the regulation of battery

charge and discharge processes, has been used in the context of our research. This methodology is especially relevant to our modeling study because we are using it as a guide to solve comparable issues in a microgrid system that incorporates solar energy sources.

The Simulink model for a hybrid wind–solar E-bike charging station includes a wind turbine, PV system, lithium-ion battery, charge controller, inverter, and DC–DC converter. The wind turbine and PV system generate power, which is input to a DC bus and connected to the battery through a charge controller. The DC bus is also connected to an inverter, which converts the power to AC power to charge the electric bike battery. The inverter output is connected to a DC–DC converter, which regulates the voltage to match the requirements of the electric bike battery. The electric bike battery is connected to a load representing the bike's charging [28]. The hybrid wind–PV microgrid power system for the proposed research that is simulated in MATLAB/Simulink is shown in Figure 3 below.

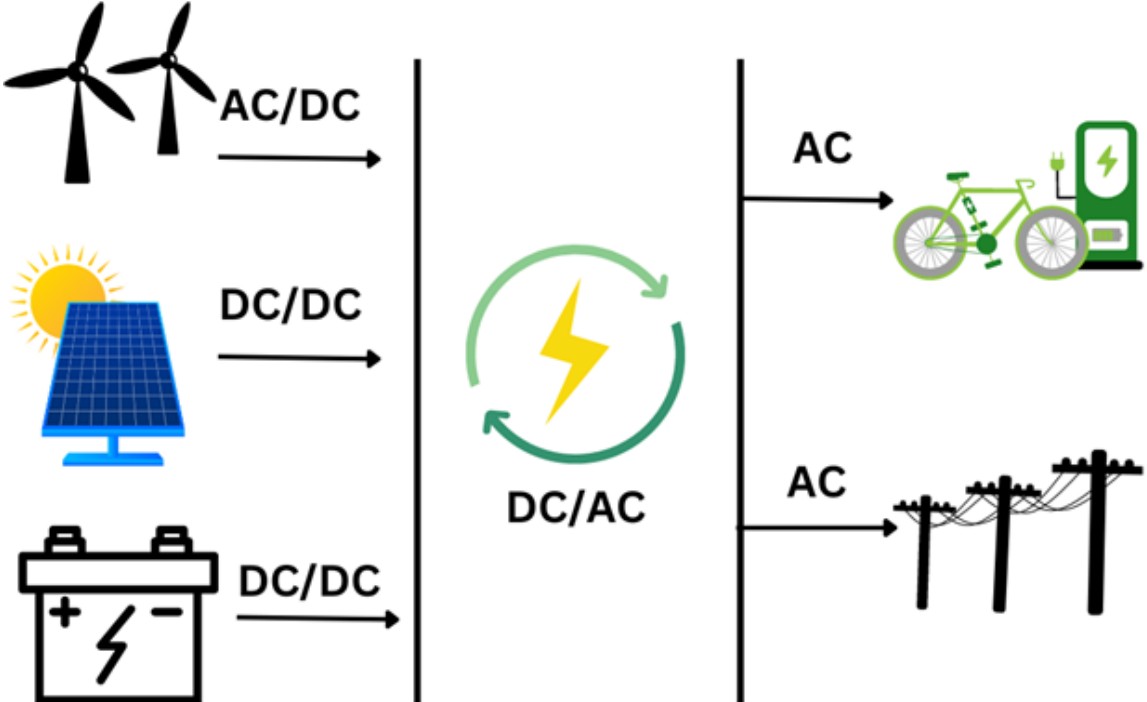

**Figure 3.** Block diagram of proposed system.

The proposed system is a block diagram comprising wind turbine (WT), PV, and energy storage systems (ESSs), such as battery banks. In this system, controllable loads include electric vehicles (EVs). The EVs are charged through DC–DC converters controlled by a charging regulation control scheme [29].

### 3.1. Wind Turbine

The wind turbines in this project have a power output of 1 kW (each) and can operate at either 24 or 48 volts. The blade rotation diameter of each turbine is 1.6 m and the radius is 0.8 m, allowing for efficient energy conversion. The turbines have a starting wind speed of 2.5 m/s, the minimum wind speed needed to generate electricity. The cut-in wind speed for these turbines is 3 m/s, indicating the wind speed at which the turbines can start producing their rated power output. These turbines are designed to withstand wind speeds up to a maximum safe wind speed of 25 m/s, ensuring their safety in high wind conditions. The outer diameter of the tower used to support the turbines is 60 mm and has a thickness of 5 mm, providing adequate support for the turbines. This project uses two turbines, each with a power output of 1 kW, for a combined total power output of 2 kW. Table 3 below contains the required parameters of the proposed wind turbine

**Table 3.** Wind turbine specs.

| Parameter | Unit | Value |
|---|---|---|
| Wind turbine rated power | Kw | 1 (each) |
| Rotor diameter | m | 1.6 |
| Cut-in speed | m/s | 3 |
| Maximum safe speed | m/s | 25 |
| Voltage | V | 24/48 |
| Outer diameter of tower | Mm | 60 |

### 3.2. Mathematical Modelling of Wind Turbine

The mathematical modelling must be examined and studied before developing the Simulink model. So, here is the mathematical model of the wind turbine. The power generated from the turbine in a wind energy conversion system could be stated numerically as:

$$Power = \frac{1}{2}\rho A v^3 C_p(\lambda, \beta) \tag{1}$$

where

- $C_p$ = aerodynamic coefficient;
- $\lambda$ = tip speed ratio (TSR) = $\omega R/v$;
- $\beta$ = pitch angle;
- $\rho$ = air density (1.22 kg/m$^2$);
- $A$ = rotor swept area (m$^2$);
- $v$ = instantaneous velocity of the wind (m/s).

Using a mathematical wind turbine model, we have modelled the power curve and TSR vs. Cp for a proposed wind turbine of 1 kW using MATLAB Simulink, illustrated in Figure 4 below:

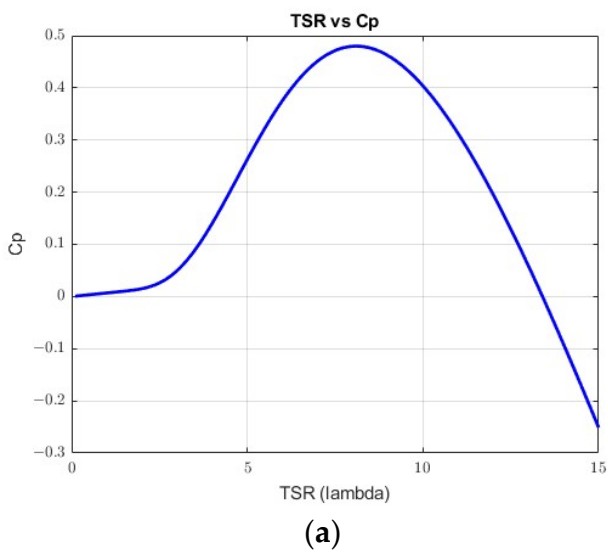

**(a)**

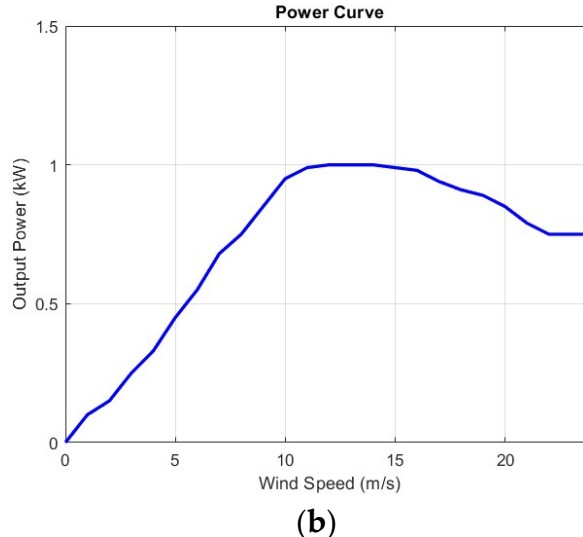

**(b)**

**Figure 4.** (**a**) TSR vs. Cp. (**b**) Power curve of 1 kW wind turbine.

### 3.3. Solar PV

The solar panel system in this research consists of four arrays of single crystal A-level solar panels. Each panel has a voltage output of 30 V and a maximum current output of 10 A, resulting in a maximum power output of 300 W per panel. The panels have a high conversion efficiency, with a minimum efficiency of 15%, ensuring efficient energy conversion from sunlight. The panels are of A-level quality, indicating high performance and durability. The four arrays of panels are designed to work in tandem, providing a combined maximum power output of 1200 W. This solar panel system is ideal for various

residential, commercial, or industrial applications where renewable energy sources are preferred. The high-quality panels are designed to withstand harsh weather conditions, ensuring longevity and reliability. Table 4 below contains the required parameters of the proposed solar PV.

**Table 4.** Solar PV specs.

| Parameter | Unit | Value |
| --- | --- | --- |
| PV power | kW | 1.2 |
| Conversion efficiency | % | 15 |
| Each panel O/P voltage | V | 30 |
| Maximum current O/P | A | 10 |

### 3.4. Mathematical Modelling of Solar PV

Modelling solar PV cells is based on the operation and features of a conventional silicon PN junction diode. The IV characteristics of a typical silicon PN junction are quantitatively provided by Equation (2), as mentioned below.

$$I = Io\left(e^{\frac{V}{Vt}} - 1\right) \tag{2}$$

where

- k = Boltzmann's constant (J/K);
- $t$ = temperature (K);
- q = electron charge in coulombs;
- $Vt$ = voltage equivalent of the temperature = kT/q;
- $Io$ = reverse saturation current (A).

### 3.5. Simulink Modelling

The proposed MATLAB/Simulink model illustrated in Figure 5 below combines actual wind and solar data for both onshore and offshore sites, allowing for a thorough analysis of the hybrid wind and solar PV energy system for the E-bike charging station. Adding DC–DC converters, a battery storage system, and an E-bike charger, which collectively simulate the functional elements of such a system, further improves this complex model. The model's ability to reflect the changing nature of wind energy generation properly due to the input of actual wind data enables a thorough analysis of how it affects the hybrid system's operation [30]. By maximizing power conversion between wind and solar sources, the DC–DC converters play a crucial role in raising the effectiveness of energy extraction. Incorporating a battery storage unit ensures the smooth absorption of extra energy during high output and the facilitation of its controlled discharge during low production. The E-bike charger demonstrates how renewable energy is used to charge electric vehicles. It is an example of a practical application of the hybrid system's output. Overall, using actual wind data, this Simulink model captures the intricacies of hybrid energy systems and offers a solid framework for assessing their potential for producing sustainable energy and for practical application.

The proposed system's simulation design has been implemented using Simulink/MATLAB. The simulation integrates the complex models of a wind turbine (Figure 6), MPPT-based PV system (Figure 7), DC–DC converters, AC–DC converters, and an E-bike charging station. The wind turbine model encompasses the aerodynamic and mechanical attributes, whereas the solar PV model replicates the energy generation by considering the prevailing environmental circumstances. The incorporation of DC–DC converters serves to optimize the transmission of power between sources and loads, whereas AC–DC converters enable the conversion of alternating energy into direct current for storage and utilization. The E-bike charging station model incorporates the charging profiles and energy demands. Using a simulation-based methodology provides a framework for evaluating the hybrid charging

station's dynamic characteristics, effectiveness, and overall operational effectiveness across various scenarios.

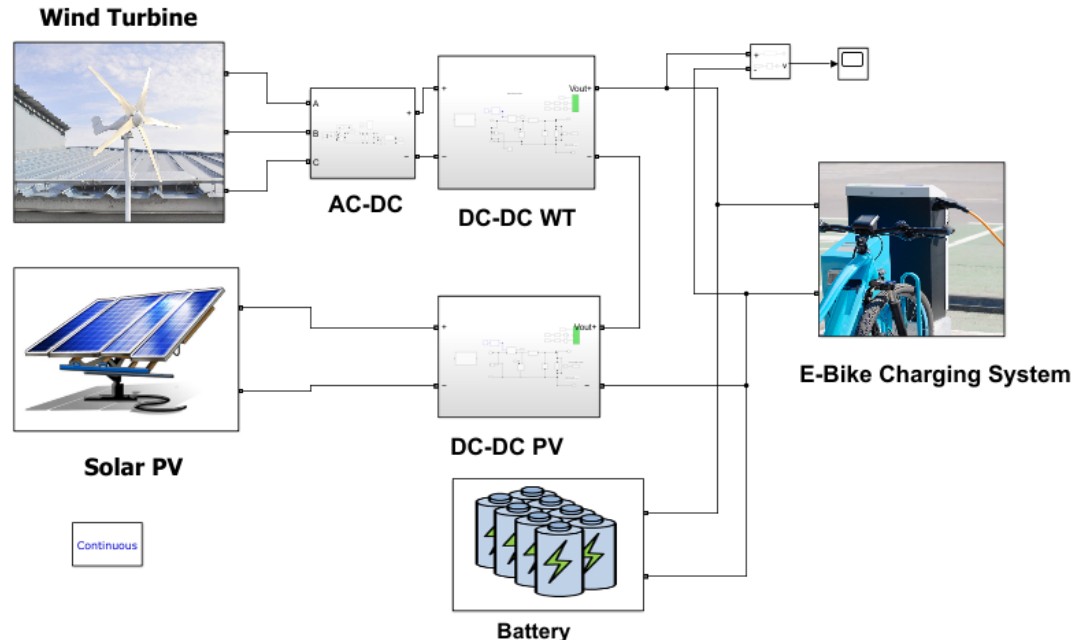

**Figure 5.** Simulink model of the proposed system.

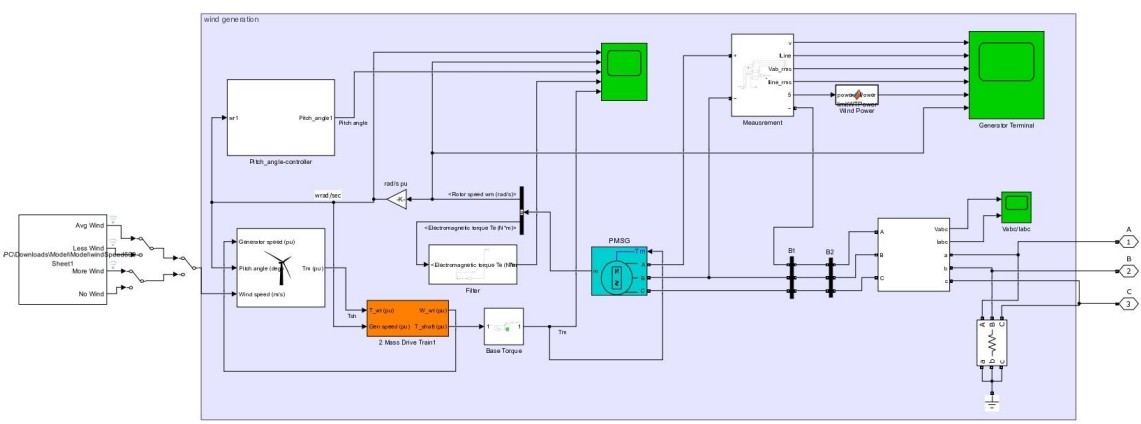

**Figure 6.** Simulink model of the proposed wind turbine.

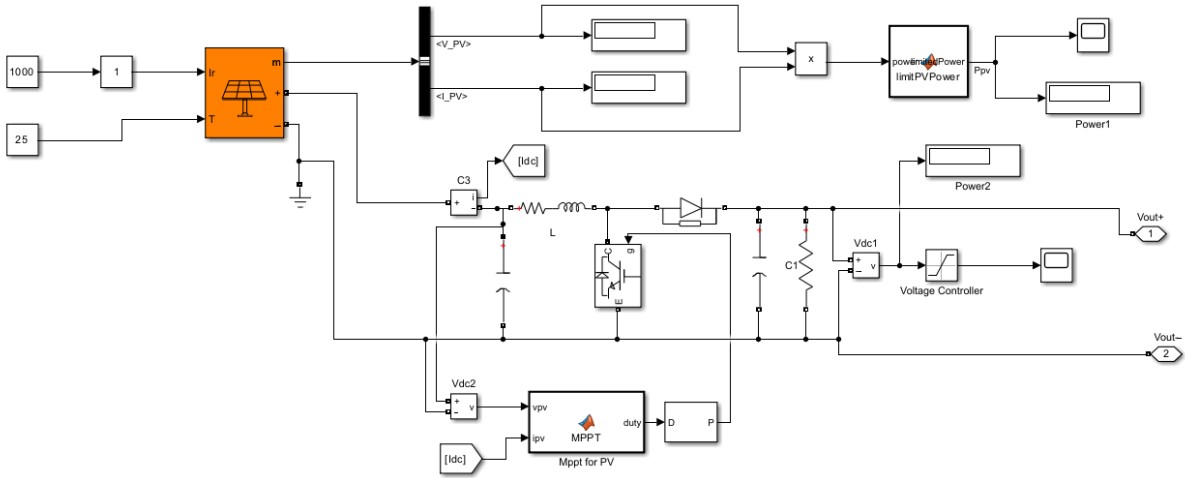

**Figure 7.** Simulink model of the proposed PV system.

### 3.6. Results & Discussions

The Simulink results of this research have important significance for the growth of renewable energy integration in urban transportation systems, as shown in Figure 8. A thorough representation of the simulation findings is shown in Figure 8, which also shows the detailed connections between the E-bike charging infrastructure, solar photovoltaics (PV), and wind turbines. A comprehensive review of the proposed wind turbine is shown in Figure 8a, covering essential factors like voltage, current, power output, and rotor speed, all of which were assessed under conditions of average wind speed. Understanding the effectiveness and viability of wind energy generation for recharging E-bikes requires an understanding of these metrics. The simulation results for the proposed solar panel system are shown in detail in Figure 8b, together with information on solar power production and voltage levels under conditions of average irradiance and temperature. These data offer important insights into the performance and efficiency of the hybrid system's solar PV component.

The simulation results are shown in Figure 8c and explain the state of charge (SOC) of the E-bike batteries during the charging process. The SOC for up to six E-bikes charging simultaneously is specifically examined using the suggested model. In order to provide effective and dependable E-bike charging operations, understanding SOC is essential. The effectiveness of wind and solar energy sources for charging E-bikes can be evaluated by using the suggested Simulink model in a variety of different scenarios. This model makes it easier to investigate different setups and operational scenarios, providing a thorough grasp of the possible uses and advantages of renewable energy integration in the context of transportation systems in cities.

These findings are in line with the general objectives of environmental sustainability and green mobility solutions, and they considerably promote sustainable transportation while lowering carbon emissions.

We used the suggested Simulink model to conduct a thorough analysis of numerous scenarios, which is shown in Table 5. This investigation focused on the impact of four different scenarios on the state of charge (SOC) and the time needed to reach complete 100% SOC for E-bike batteries. Each scenario included elements of wind energy, solar energy, and lithium-ion battery technology.

**Table 5.** Comparison of different cases.

| Cases | SOC% Increase in 1 s | Time for 100% SOC |
|---|---|---|
| Max wind, no PV, no battery | 0.4% | 250 s |
| No wind, max PV, no battery | 0.35% | 285 s |
| Max wind, max PV, no battery | 0.5% | 200 s |
| Max wind, max PV, Li-battery (48 V) | 7% | 15 s |

The following table provides a clear comparison of these cases:

Case 1: Maximum wind, no PV, no battery: In this setup, which just utilizes wind power and lacks support from solar power or batteries, the SOC shows a negligible rise of 0.4% in just one second, while it takes over 250 s to reach complete 100% SOC.

Case 2: Maximum PV, no battery, no wind: The goal in this scenario is to utilize solar energy to the fullest extent possible without the help of wind power or a battery. Within one second, the SOC increases marginally by 0.35%, and it takes roughly 285 s to reach its maximum value.

Case 3: Maximum wind, maximum PV, no battery: In this configuration, maximum wind and maximum solar energy are used, but no battery storage is used. Within one second, the SOC increases by 0.5%, and it reaches 100% in about 200 s.

Case 4: Maximum wind, maximum PV, Li-ion battery (48 V): In this case, a 48 V lithium-ion battery is used in addition to maximum wind and maximum solar energy. Here, the SOC increases significantly by 7% in just one second, and after a remarkable 15 s, it reaches a stunning 100% SOC.

These results highlight the important role that battery integration plays, particularly in Case 4, where the use of a lithium-ion battery produces a charging process that is noticeably quick, lowering the time needed to reach a full SOC and increasing the overall effectiveness of the E-bike charging system. This analysis offers insightful information about the efficacy of various energy configurations for charging e-bikes, assisting in the improvement of efficient and sustainable charging solutions.

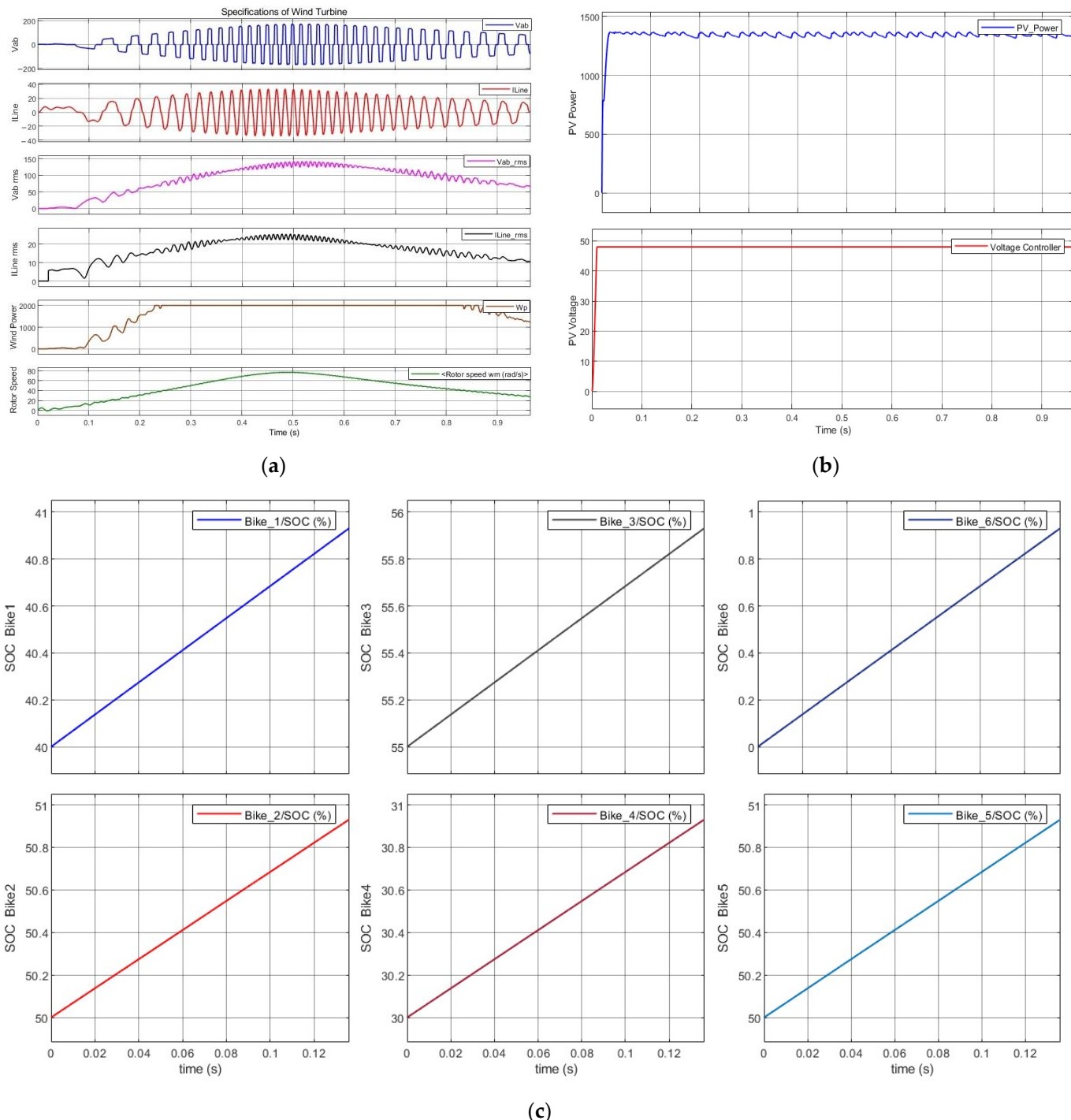

**Figure 8.** (**a**) Simulation results of wind turbine. (**b**) Simulation results of solar PV. (**c**) Simulation results of SOC% of E-bike charging.

### 3.7. AEP Analysis of Wind/Solar

The AEP of wind and solar energy has been thoroughly examined in this research article for both onshore and offshore locations. It has been determined via careful analysis of wind resources' performance parameters and energy outputs that offshore sites have a significantly higher AEP than their onshore counterparts, illustrated in Figure 9 for offshore and Figure 10 for onshore locations. Similarly, the AEP analysis is calculated for PV solar for both onshore and offshore sites, illustrated in Figure 11. The study considers several variables: wind patterns, solar irradiation levels, geographic location, and technology developments in onshore and offshore installations [26]. The enormous potential of offshore locations to collect renewable energy on a large scale is highlighted by the observed considerable differential in AEP, which has positive implications for sustainable energy generation and climate mitigation measures.

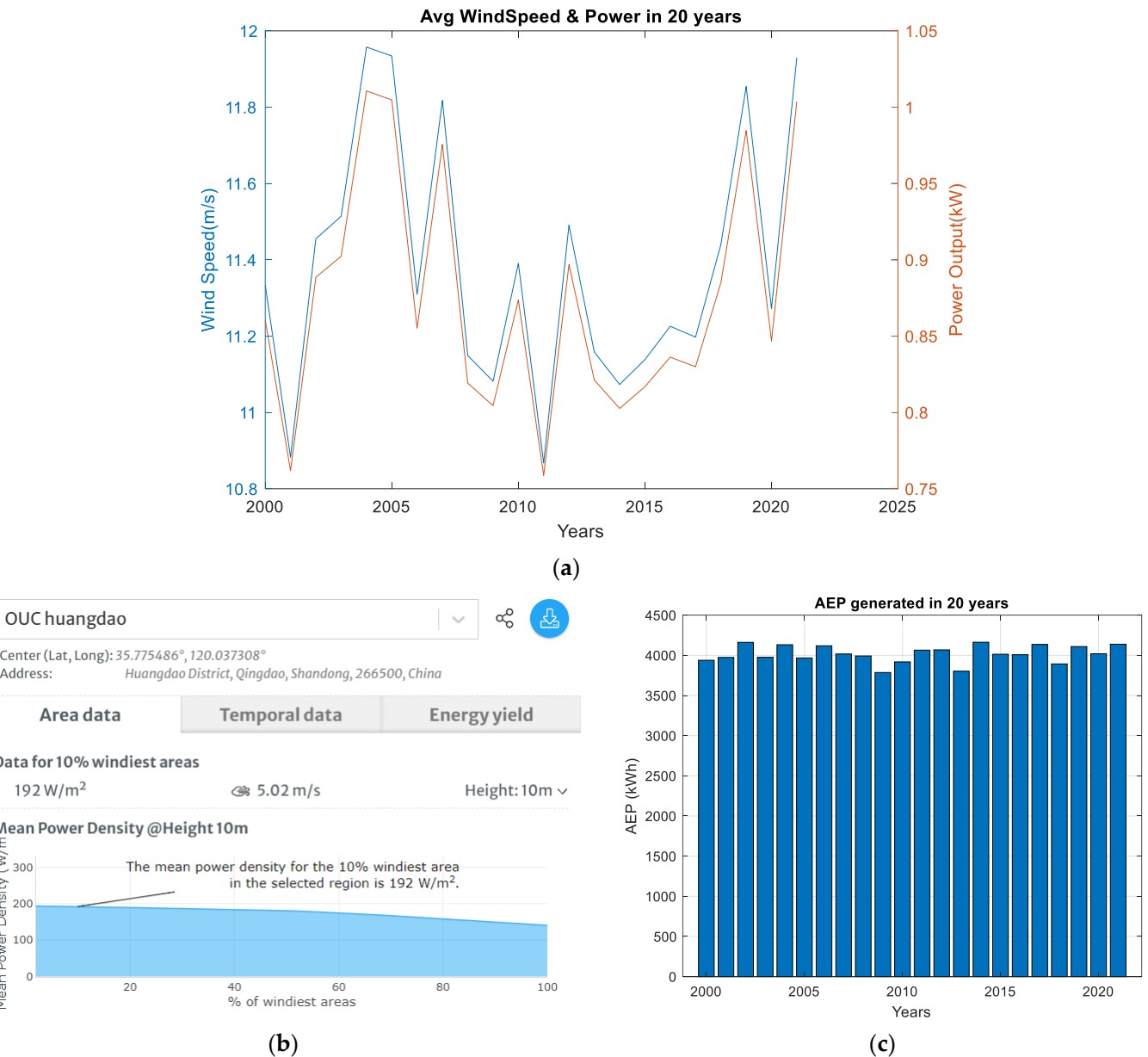

**Figure 9.** (**a**) Average wind speed and power output in Huangdao over 20 years. (**b**) Wind data available at Huangdao (offshore). (**c**) AEP calculation over 20 years.

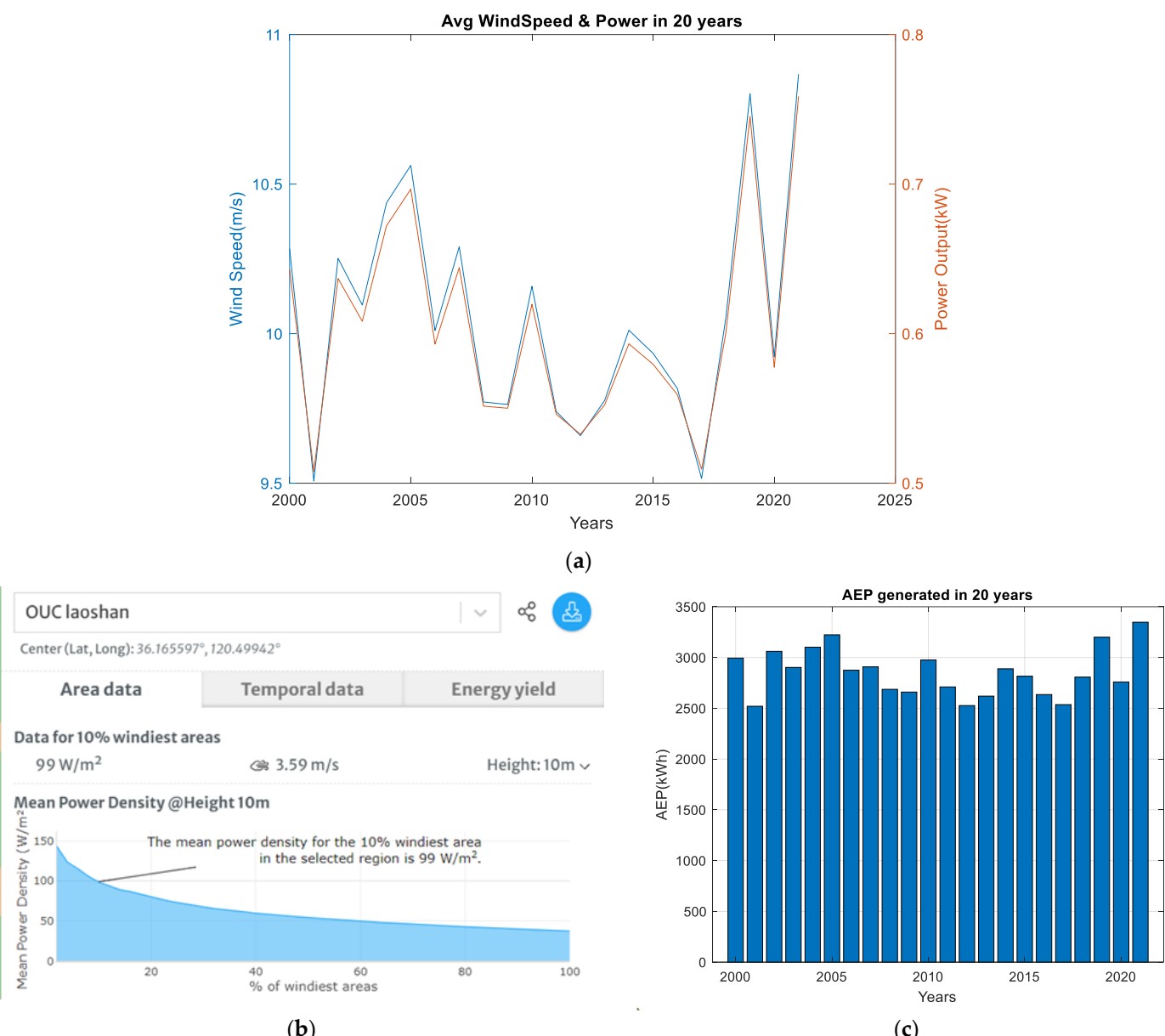

**Figure 10.** (**a**) Average wind speed and power output in Laoshan over 20 years. (**b**) Wind data available at Laoshan (onshore). (**c**) AEP calculation over 20 years.

This research examines the actual use of the estimated energy values and the comparative analysis of wind and solar AEP for onshore and offshore sites. In particular, the research examines how to compute the E-bike charging capacity using the calculated AEP data. The study provides insightful information on the potential for environmentally friendly transportation options by considering the energy requirements for charging electric bicycles and comparing them to the AEP statistics. The results show that E-bike charging stations at offshore sites are more feasible because of the higher AEP values. This integration highlights the many advantages of using renewable energy and shows a practical way to cut carbon emissions in the transportation industry. The comprehensive methodology emphasizes offshore locations' excellence in producing renewable energy and shows how they directly support environmentally beneficial transportation options.

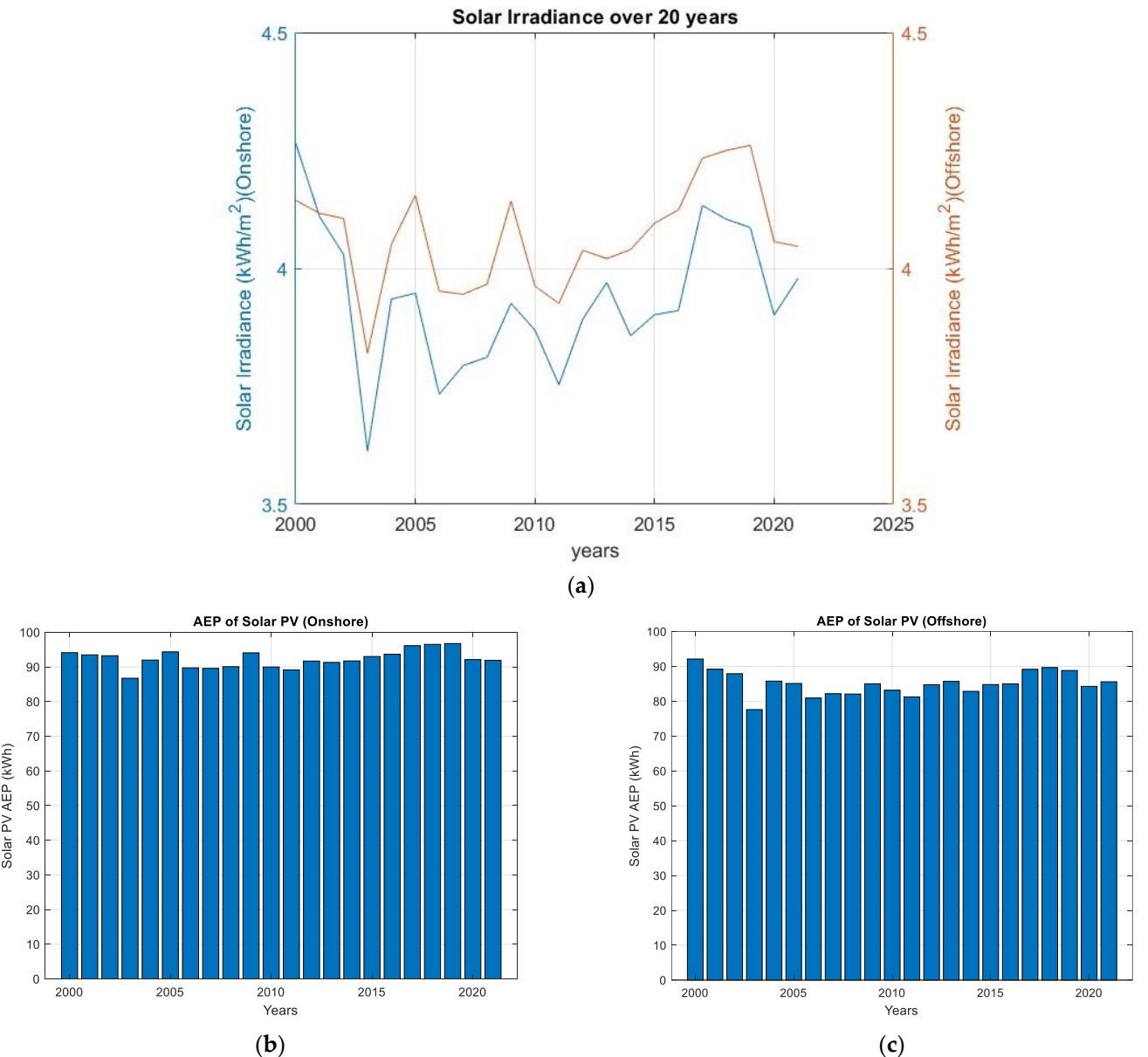

**Figure 11.** (**a**) Solar irradiance in Laoshan and Huangdao over 20 years. (**b**) AEP of solar PV in Laoshan over 20 years. (**c**) AEP of solar PV in Huangdao over 20 years.

## 4. E-Bike Calculations

### 4.1. E-Bike Calculation for Case 1—Onshore (Laoshan)

Using the designed system, we have made some assumptions and calculations for E-bike in OUC to calculate the number of E-bikes charged per day and year.

These are the following assumptions:

- The electric bike charger has a power rating of 400 watts.
- The charging time required to charge one E-bike from the available power is 4 h.
- The battery voltage is 48 V, and the capacity of each battery block is 4.8 kWh (100 Ah × 48 V/1000).
- The charging efficiency is assumed to be 90% for wind and solar power.

With these assumptions, we can calculate the available energy from each source per day:

- Wind turbine: 2988 kWh/year/365 days = 8.2 kWh/day;

- Solar PV: 96 kWh/year/365 days = 0.27 kWh/day;
- Lithium battery: 4.8 kWh.

Next, we can calculate the hybrid power available per day by adding up the energy from each source:

Hybrid power available per day (on average) = wind power + solar power + battery power = 2 × 8.2 kWh/day + 0.27 kWh/day + 4.8 kWh/day = 22 kWh/day.

Assuming a charging time of 4 h per E-bike (as we know, fully discharged lithium-ion E-bike batteries require 3.5 to 6 h to recharge), we can calculate the number of E-bikes that can be charged per day and year:

Number of E-bikes charged per day = hybrid power available per day/(charging time per E-bike × charger power rating) = (22 kWh/day)/(4 h × 500 watts).

Number of E-bikes charged per year = 11 E-bikes per day × 365 days = 4015 E-bikes per year on average based on AEP.

### 4.2. E-Bike Calculation for Case 2—Offshore (Huangdao)

With the assumptions above in Case 1, the calculations for Case 2 are given below:

- Wind turbine: 4045 kWh/year/365 days = 11 kWh/day;
- Solar PV: 100 kWh/year/365 days = 0.28 kWh/day;
- Lithium battery: 4.8 kWh.

Next, we can calculate the hybrid power available per day by adding up the energy from each source:

Hybrid power available per day = wind power + solar power + battery power = 2 × 11 kWh/day + 0.28 kWh/day + 4.8 kWh/day = 27.08 kWh/day.

Assuming a charging time of 4 h per E-bike (as we know, fully discharged lithium-ion E-bike batteries require 3.5 to 6 h to recharge), we can calculate the number of E-bikes that can be charged per day and year:

Number of E-bikes charged per day = hybrid power available per day/(charging time per E-bike × charger power rating) = (27.08 kWh/day)/(4 h × 500 watts) ≈ 14 E-bikes per day on average. Number of E-bikes charged per year = 14 E-bikes per day × 365 days = 5110 E-bikes per year on average based on AEP.

## 5. Cost of Energy

The present study utilized economic analyses based on the cost of energy (COE), a widely used and accepted metric. COE is a standard method for evaluating the cost-effectiveness of different energy sources and technologies by considering the total cost of generating electricity over the lifetime of a project and dividing it by the total electricity generated. Typically, COE calculations account for capital costs, operating and maintenance expenses, fuel costs, and the project's expected lifetime. These calculations aim to provide a standardized and comparable measure of the cost of electricity production across different energy sources. The cost analysis of the proposed project is given below:

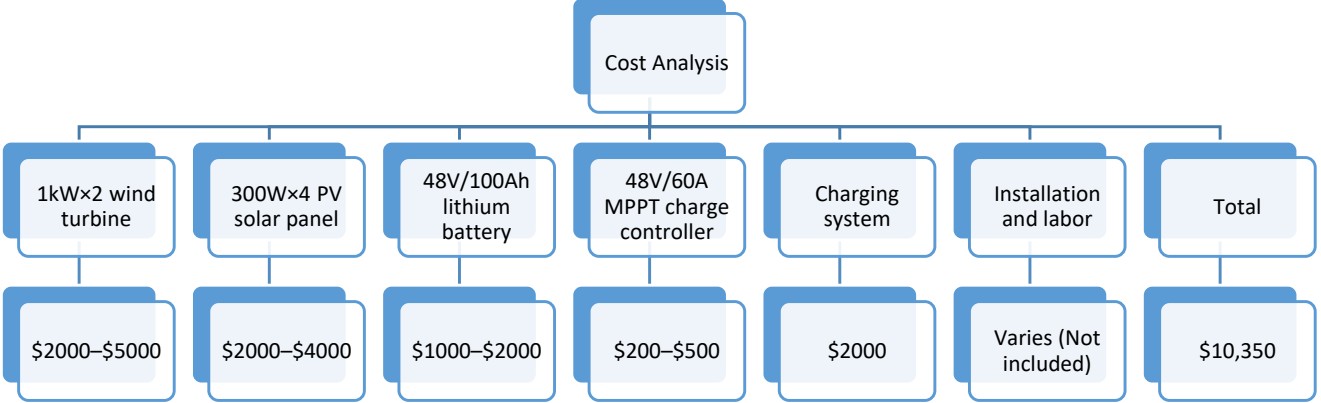

To calculate the COE of the proposed project, we need to consider the project's total cost, the equipment's lifetime, and the energy generated over the project's lifetime.

$$COE = \frac{NPV \ of \ total \ cost \ of \ Energy}{NPV \ of \ Electricity \ produced}$$

$$COE = \sum \frac{(I_t + M_t + F_t)}{(1 + r)^t} / \sum \frac{(E_t)}{(1 + r)^t}$$

The COE is a widely used metric for evaluating the economic viability of power-generating assets. The COE is calculated by taking the net present value (NPV) of the total costs associated with building and operating the asset and dividing it by the total electricity generation over the system's lifetime. The costs considered in the calculation include the initial investment expenditures ($I$), which encompass the upfront capital costs of constructing the power-generating asset and maintenance and operation expenditures ($M$), covering ongoing operational expenses. If the power generation relies on fuel, such as in fossil fuel-based plants, fuel expenditures ($F$) are also considered. On the other side of the equation, the total output of the asset includes the sum of all electricity generated ($E$) over its operational lifespan.

Additionally, the COE equation incorporates two critical factors: the discount rate of the project ($r$), which accounts for the time value of money and reflects the project's cost of capital, and the life of the system (n), representing the expected operational lifespan of the asset. In Case 1, COE for the onshore site of the proposed project is about \$0.62/kWh, and in Case 2, for the offshore site, COE is \$0.46/kWh. These analyses are done by using the AEP of both sites.

The above results and calculations show a brief comparison between offshore and onshore sites, illustrated in Table 6 below. The physical locations of offshore and onshore sites are illustrated below in Figure 12.

**Table 6.** Comparison between onshore and offshore sites.

| Parameters | Onshore (Laoshan) | Offshore (Huangdao) |
|---|---|---|
| AEP | 2988 kWh/year | 4045 kWh/year |
| Average wind speed | 8.0561 m/s | 11.3852 m/s |
| Average solar irradiance | 4.4727 kWh/m$^2$ | 4.3261 kWh/m$^2$ |
| AEP (solar) | 96 kWh/year | 100 kWh/year |
| E-bike charged on average | 4015 E − bikesperyear | 5110 E − bikesperyear |
| COE | \$0.62/kWh | \$0.46/kWh |

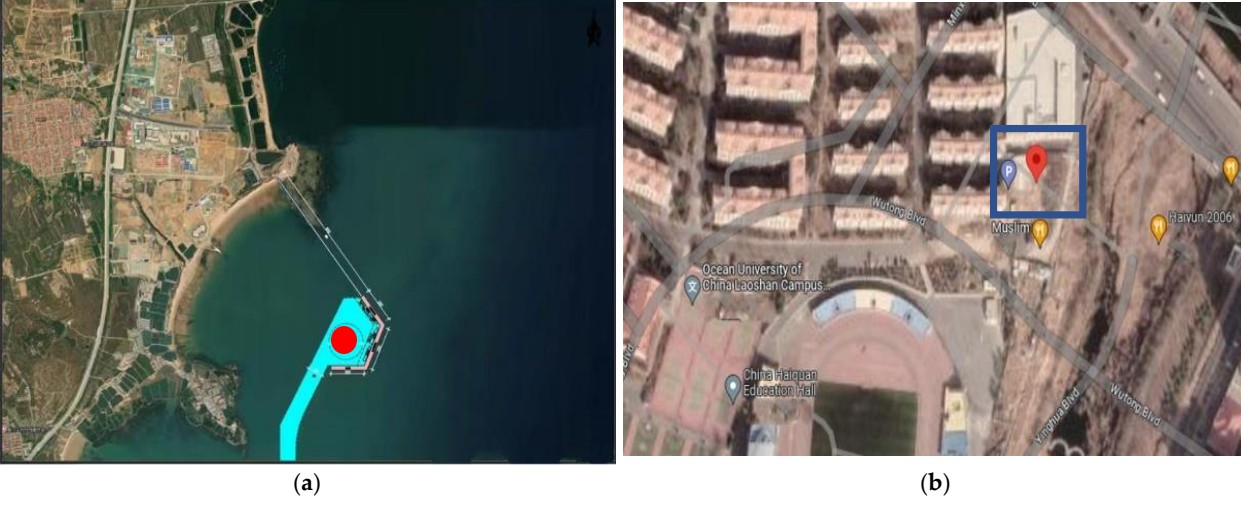

(**a**)　　　　　　　　　　　　　　　　　　(**b**)

**Figure 12.** (**a**) Offshore site in Huangdao. (**b**) Onshore location in Laoshan.

## 6. Future Scope

Future hybrid wind and solar E-bike charging station technology has considerable potential for revolutionary breakthroughs. Using hydrogen-based energy storage systems instead of lithium-ion batteries is one intriguing option that might considerably improve energy efficiency, storage capability, and sustainability. Furthermore, incorporating Internet of Things (IoT) technology to enable wireless charging marks a revolutionary advancement in the industry. The combination of IoT-enabled wireless charging will not only do away with the requirement for physical connectors but also speed up the charging procedure, making it more practical and available for E-bike users. A revolution in clean, effective, and user-friendly E-bike charging infrastructure could result from the convergence of lithium-ion batteries with IoT wireless charging as technology advances, opening the way for a greener and more connected world. In the onshore location, there are two buildings near the installation of wind turbine, so we were required to perform micro sitting for the purpose of building blockage effect.

Future research and development could greatly benefit from addressing the key issue of Li-ion battery overheating and overcharging in E-bikes. As the use of E-bikes increases, it is crucial to guarantee the security and dependability of their energy storage systems. The deployment of sophisticated controllers and the creation of rigid safety standards can be the key areas of future research in this field. Advanced algorithms for real-time monitoring and control, the detection of potential overcharging or overheating circumstances, and the implementation of preventative steps to reduce hazards might be incorporated into these controllers. In order to prevent accidents and guarantee the durability of these batteries, safety standards for Li-ion batteries used in electric bicycle applications must be developed and adhered to. This future research improves the security of E-bike riders while simultaneously advancing the ethical and sustainable growth of electric mobility options.

## 7. Conclusions & Recommendation

Hybrid wind and PV E-bike charging stations are promising solutions for sustainable transportation and reducing carbon emissions. The system's size and capacity, based on E-bikes' charging requirements and location, are crucial for maximizing energy output. Qingdao's climate and wind potential suit a hybrid charging station. Although the cost of building a hybrid system can be high, long-term savings in energy costs and environmental benefits make it a worthwhile investment. Further research is necessary to optimize their design and performance. In this research report, we have calculated the number of E-bikes that can be charged daily and yearly using a hybrid power system consisting of a wind turbine, solar PV, and a lithium battery for two locations (onshore and offshore). We have analyzed the wind and solar properties for the installation's location using the data of the last 20 years and calculated the AEP for wind turbines and solar PV. We have calculated that the hybrid power available is 27.08 kWh/day offshore and 22 kWh/day onshore. This research study concludes that in the case of offshore, 5110 E-bikes can be charged per year and in the case of onshore, 4015 E-bikes can be charged per year on average based on AEP. We have also calculated the COE for 20 years for the proposed project, which is $0.62/kWh onshore and $0.46/kWh offshore. In a nutshell, the offshore model is a good option to install the proposed project in terms of AEP and COE as we can generate more power in offshore. However, it is essential to note that the actual charging capacity may vary depending on various factors such as weather conditions, battery storage capacity, and charger power rating, among others.

It is clear from a detailed analysis that an offshore location is the most appropriate option for the installation of the hybrid system. The research includes an evaluation of the annual energy production (AEP) produced from both wind and solar sources. When compared to its onshore counterpart, the offshore location repeatedly showed greater potential for energy generation. Offshore areas typically benefit from greater, more consistent wind speeds, which increases the amount of electricity produced by wind turbines. Offshore wind and solar resources working together can offer a more consistent and balanced elec-

tricity supply all year long, allaying concerns about intermittent energy. Offshore locations provide enough room for the installation of both wind and solar components, enabling effective resource exploitation. Scalability can be considered while designing offshore systems, making it simpler to increase capacity in response to rising demand. The ecological impact of offshore projects can be reduced with careful planning and environmental concerns.

In conclusion, it is recommended that the hybrid wind/PV E-bike charging station should be built offshore due to the area's better potential for energy generation, which will lead to a more effective and sustainable system. This decision supports the larger objective of giving E-bike charging stations a dependable and environmentally beneficial energy supply.

**Author Contributions:** Conceptualization, Y.X.; methodology, W.A.; software, W.A.; validation, W.A., Y.X. and L.-Y.Z.; formal analysis, Y.X. and W.A; investigation, Y.X.; resources, W.A., Y.X. and G.-Z.C.; data curation, W.A.; writing—original draft preparation, W.A.; writing—review and editing, W.A.; visualization, W.A., and Y.X.; supervision, Y.X. All authors have read and agreed to the published version of the manuscript.

**Funding:** This research was funded by the special fund for offshore wind power intelligent measurement and control research center and laboratory construction at Ocean University of China.

**Institutional Review Board Statement:** Not applicable.

**Informed Consent Statement:** The software employed in the research is openly available in https://www.mathworks.com/products/new_products/latest_features.html (accessed on 14 September 2023).

**Data Availability Statement:** Not applicable.

**Acknowledgments:** This research is supported by Ocean University of China.

**Conflicts of Interest:** The authors declare no conflict of interest.

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
