# Peer review of "Hybrid Wind/PV E-Bike Charging Station: Comparison of Onshore and Offshore Systems"

_sustainability, doi:10.3390/su152014963_

Round 1
Reviewer 1 Report
In this manuscript, the authors researched hybrid Wind and PV E-bike charging stations for the sustainable transportation. The size and capacity of the system on E-bikes' charging requirements and location are important for maximizing energy output. They determined the number of E-bikes using a hybrid power system consisiting of a wind turbine, solar PV and a lithium battery in onshore and offshore. This work is helpful to hybrid power supply. However, before it can be published, the following issues should be concerned carefully.
1. A picutre of real charging station should replace Figure 1 which is only an imaginary one.
2. The steps for a solar-wind hybrid EV charging station should be Figure 2. It lacks figure caption.
3. These words in Fig.4 and Fig.5 are not clear at all.
Author Response
Dear Reviewer,
Please see the attachment below to find out the point-by-point response to the comments.
Regards

Reviewer 2 Report
Comments
In this work, the authors reviewed the progress of research on Hybrid Wind/PV E-Bike Charging Station: Comparison of Onshore and Offshore System. In my view, the following issues need to be addressed before consideration for publication.
1. Please add your own diagram of the mechanism section.
2. Please provide some of your own thinking and outlook.
3. Please comment on the previous work what can be improved.
Minor editing of English language required.
Author Response

(The authors gave the same response as above.)

Reviewer 3 Report
In this manuscript, the authors presented the hybrid wind/PV E-bike charging station: comparison of the onshore and offshore systems. The research work represents scientific interest. The manuscript can be improved if the authors can revise the following points:
1. Avoid employing multiple citations for a single point; such as [9–12], [19–22].
2. Please enhance the readability of this manuscript by providing a comprehensive explanation within the Results and Discussion section.
3. In Figure 7, there should be a representation of the X and Y axes; Proper arrangement and reduce the line thickness of the graph.
4. The authors should make a recommendation regarding which system is suitable.
Minor editing of the English language is required.
Author Response

(The authors gave the same response as above.)
